# Successful Treatment of Complicated Influenza A(H3N2) Virus Infection and Rhabdomyolysis with Compassionate Use of IV Zanamivir

**DOI:** 10.3390/ph16010085

**Published:** 2023-01-07

**Authors:** Maren Alchikh, Patrick E. Obermeier, Brunhilde Schweiger, Barbara A. Rath

**Affiliations:** 1Vaccine Safety Initiative, 10437 Berlin, Germany; 2Laboratoire Chrono-Environnement, Université Bourgogne Franche-Comté, 25030 Besançon, France; 3National Reference Centre for Influenza, Robert Koch-Institute, 13353 Berlin, Germany

**Keywords:** influenza A(H3N2), multiorgan failure, ECMO, hemofiltration, intravenous zanamivir

## Abstract

In 2019, EMA licensed intravenous (IV) zanamivir for severe influenza virus infection in children over 6 months as well as adults. Prior to that, it was possible via a compassionate use program. We present successful compassionate use of IV zanamivir in a 14-year-old female with severe influenza A(H3N2) and multi-organ failure, who had failed oral oseltamivir. Her illness was complicated by acute respiratory distress syndrome and rhabdomyolysis requiring extracorporeal membrane oxygenation and hemofiltration. Considering the broad safety margins with neuraminidase inhibitors, an adult dose of 600 mg IV BID was administered in this 60 kg patient. Influenza virus was cleared rapidly and undetectable on day 13. Creatine kinase (CK) values were dropping from 38,000 to 500 within nine days. Given the recent licensure of IV zanamivir, multi-center prospective observational studies in pediatric Intensive Care Unit patients would be beneficial to guide the most appropriate use of IV zanamivir in this vulnerable age group.

## 1. Introduction

Severe influenza virus infections may occur in healthy children with and without previously established risk factors [1,2,3]. This means the clinician must respond as soon as possible when a patient’s illness seems to be progressing. Neuraminidase inhibitor (NAI) treatment has been shown to be effective at risk of severe complications and mortality if started within 2 days of symptom onset, in some instances even beyond that time frame [4]. For patients in the Intensive Care Unit (ICU) with multi-organ failure and limited gastric motility however, oral oseltamivir may not be absorbed adequately.

For children with severe and progressive influenza disease who have failed 5 days of per os (po) oseltamivir treatment, there are limited intravenous (IV) antiviral treatment options available. IV peramivir has been licensed in Japan, South Korea and the United States [5,6]. IV oseltamivir was available for adults and children over 1 year of age, as compassionate use program from 2010 to 2013 [7]. From 2010 to 2019, IV zanamivir compassionate use was the only IV option [8]. (IV) zanamivir has since been licensed in 2019 by the EMA, with a special indication for severe influenza in adults and children over 6 months of age [9]. There are few studies of peramivir—but even less studies of IV zanamivir—in the pediatric age group.

To support the growing body of evidence, we provide a detailed description of compassionate (pre-licensure) use of IV zanamivir [8] in a previously healthy 14-year old ICU patient with severe influenza A(H3N2) virus infection, rhabdomyolysis and multi-organ failure requiring organ replacement therapy: extracorporeal membrane oxygenation (ECMO) and continuous venovenous hemodialysis (CVVHD). The day-to-day monitoring included standardized clinician-reported clinical outcomes via mobile app, virus load assessments at the National Reference Centre for Influenza and the use of creatine kinase (CK) and myoglobin levels as biomarkers for influenza-associated rhabdomyolysis.

## 2. Results

In this report, we provide a detailed account of IV zanamivir therapy in a previously healthy 14-year-old Caucasian female (60 kg, Body Mass Index 23.4 kg/cm^2^) who was transferred from an outside hospital to the adult ICU due to need of ECMO therapy. At the time of presentation to the ICU, her influenza virus infection (initially untyped) had rapidly progressed to severe disease with ARDS, circulatory instability, severe pancytopenia, rhabdomyolysis, and renal failure. The patient had failed a previous seven day extended course of oral oseltamivir. The patient was previously healthy and had no known cardiopulmonary risk factors, except for a nicotine habit.

### 2.1. Initial Presentation Prior to Treatment

#### 2.1.1. Clinical Presentation

In March 2015, the patient presented with fever, sore throat, vomiting and listlessness/lack of initiative. The next day, neurologic symptoms as dizziness, disorientation and incontinence appeared. The illness further developed to severe septic shock with acute respiratory insufficiency/ARDS (Figure 1), hyperpyrexia up to 42.5 °C and massive pancytopenia.

After rapid progression to respiratory failure, the patient was transferred to the Adult ICU on day four of illness for ECMO treatment.

#### 2.1.2. Virology

At the outside hospital, the patient had a laboratory-confirmed diagnosis of influenza. A virus infection (via multiplex PCR), but no differentiation/further amplification had reportedly been possible. On day eight of illness, the initial bronchoalveolar lavage (BAL) specimen was confirmed positive for influenza A(H3N2) by PCR at the National Reference Centre for Influenza at the Robert Koch-Institute. Follow-up testing on day twelve was once again positive for Influenza A(H3N2) with a ct value of 33. Follow-ups after this date remained negative. No other viruses were detected (tested for human rhinovirus, human respiratory syncytial virus, human adenovirus and human metapneumovirus).

#### 2.1.3. Biomarkers

The creatine kinase level was 354 U/L on day four of illness, when the patient was transferred to Charité hospital under po oseltamivir therapy. On the next day, it had increased tenfold to 3473 U/L. From day nine to ten of illness, it increased from 8019 U/L to 38,739 U/L before IV zanamivir was started.

Myoglobin was on 400 µg/L on day four of illness. It increased to 29,002 µg/L on day ten of illness, when IV zanamivir was started.

### 2.2. Treatment Decision

At the time of decision in favor of compassionate use IV zanamivir treatment, the patient had already failed a seven-day course of oral oseltamivir treatment (75 mg BID) which had reportedly been started on day three of illness. On day four of illness, extracorporeal membrane oxygenation was started. On day six, CVVHD was initiated.

On day ten of illness, the patient developed massive rhabdomyolysis with maximum CK of 38,739 U/L and myoglobin of 29,002 µg/L. As a consequence of rhabdomyolysis, history of hypoxemia and overall disease progression, she developed renal failure and increasing inflammatory parameters such as c-reactive protein of 336.4 mg/L, procalcitonin >100.0 µg/L and leukocytes 12.65/nL.

The team decided to initiate compassionate use intravenous zanamivir treatment, suspecting that the rhabdomyolysis might be directly related to the influenza virus infection [10,11,12,13,14,15,16,17].

The patient showed significant edema and compartment syndrome, her body weight had increased to 60 kg at the time. On day six, she had been started on CVVHD, which may hasten wash-out effects for NAI [18,19]. Despite a history of renal failure with elevated retention parameters (creatinine 2.30 mg/dL, Urea 70 mg/dL, potassium 4.8 mmol/L on day six of illness), the regular adult dose of 600 mg IV zanamivir BID was started on day ten (through day fifteen) to reach maximum efficacy as swiftly as possible.

The disease progression, treatment and virology data are outlined in detail in Figure 2.

### 2.3. Treatment Monitoring and Response

#### 2.3.1. Clinical Response and Disease Severity

Disease severity was measured daily using the ViVI Disease Severity Score, a 22-items-weighted composite clinical score via mobile app (‘ViVI ScoreApp’) [20]. The score consists of nine items DSU: disease severity, uncomplicated, weighed single-fold and 13 items DSC: disease severity, complicated, weighed threefold. ViVI Score values may range from 0–48 [21].

The peak (ViVI Disease Severity Score of 31, >99 percentile) was reached on day three of illness, when the patient was diagnosed with shock [21]. Under 7 days of oral oseltamivir therapy, ViVI-Scores had decreased slightly, from 31 to 27. During 5 days of IV zanamivir therapy, severity scores dropped further to a value of 26 (DOI#15) (Figure 3).

The patient stabilized gradually after completing 5 days of IV compassionate use therapy. Circulatory support with catecholamines could be phased out on DOI#30. Due to ARDS with extensive oxygenation problems, extracorporeal membrane oxygenation was continued for 50 days. It was finally possible to wean the patient off ventilation for 6 h per day until the discharge to a rehabilitation clinic on day 65.

#### 2.3.2. Virologic Response

Influenza A(H3N2) virus was detectable in BAL specimen on day eight of illness (DOI#8) and on day twelve of illness (DOI#12, ct 33), after zanamivir therapy had been started. Virus load had become undetectable by RT-PCR on day thirteen and remained undetectable thereafter. Resistance testing was not feasible from diluted swabs.

#### 2.3.3. Biomarker Response

Within three days, the rhabdomyolysis decreased rapidly with myoglobin level from 29,002 µg/L on DOI#10 to 4437 µg/L on DOI#16 (Figure 4).

CK levels normalized within 20 days with CK dropping from CKmax 38,739 U/L on DOI#10, CK = 5186 U/L on DOI#15, to 152 U/L on DOI#20 (not shown) (Figure 5).

## 3. Discussion

We report on successful compassionate use IV zanamivir therapy in a 14-year-old Caucasian female with severe influenza A(H3N2) virus infection, acute respiratory failure, ARDS, and rhabdomyolysis requiring ECMO and CVVHD. PO oseltamivir failed to eliminate the virus. The decision to start compassionate-use IV zanamivir treatment was based on continued disease progression with multi-organ failure and rhabdomyolysis. Determining the most appropriate zanamivir in multi-morbid pediatric ICU patients is challenging, despite the wide safety margin of the drug. Individual-level pharmacokinetics are difficult to predict, considering altered protein binding and edema during ECMO therapy, which also affects volume of distribution and protein binding [22]. Potential drug–drug interactions may incur with other medications used in ICU patients, in addition to potential renal failure and/or wash-out effects in patients with continuous veno-venous hemodiafiltration (CVVHD). Other challenges include the accurate and measurement of clinical endpoints in children, consistency in clinical patient follow-up given the rapid staff turnover in the ICU and transfer between wards and institutions and the inability to obtain patient-reported outcomes in intubated patients.

In this study, we evaluated the exact level of improvement/deterioration during therapy using a clinician reported outcome measure, the ViVI ScoreApp, i.e., a composite clinical disease severity score that was previously validated in children and adults [20]. The digital format via mobile app, audit trails and time stamps hold clinical staff accountable to report disease severity regularly and in a consistent manner. In our case, we collected the disease severity score daily with the ViVI-Score App at the bedside during the entire hospital stay. Standardized disease severity measurements such as the ViVI-ScoreApp, may be helpful to monitor and quantity overall clinical improvement under therapy—the bedside assessment via mobile application can be completed within 3 min by ICU staff.

Adequate dosing is of critical importance, especially in patients with life-threatening illness and possible NAI resistance. Resistance testing however, takes significant time and is often not available at the time of emergency decision making. Access to pharmacokinetics was not available, creating the need for treatment without knowing these two key parameters: viral resistance and drug levels. The assumption was made that po oseltamivir might not have been adequately absorbed in this patient due to gastroplegia, reflux and diarrhea [23,24,25]. In their case reports, Jahns et al. and Mazzitelli et al. have described the excellent outcome of three adults with influenza and myocarditis with gastroplegia received IV zanamivir [26,27]. Suboptimal drug levels may have increased the risk of viral resistance to oseltamivir [28,29]. The impact of ECMO and CVVHD on drug clearance was uncertain, and lack of access to rapid-turnaround pharmacokinetics further challenges the selection of the most adequate dose. ECMO may lead to significant changes in pharmacokinetics of NAI in three ways: drug sequestration by the circuit, increased volume of distribution and altered drug clearance [22]. CVVHD may lead to wash out effects [18,19]. Pediatric dosing was not established at the time, and significant dosing reduction was recommended for patients with renal failure [8]. We decided to treat the patient with the regular adult (non-renal failure) dose of 600 mg (20 mg/kg) BID for the following reasons: the patient’s weight was close to adult size (60 kg), zanamivir was known to exhibit a wide safety margin [30,31], ECMO and edema both increase volume of distribution and CVVHD may speed up drug filtration and clearance. To establish individualized evidence-based treatment regimens in children and adolescents with severe influenza infection, it would be desirable to have access to specialized laboratories with capacity to measure rapid turnaround PK, virus load and resistance testing. That way, it would be possible establish optimized timing and dosing of IV zanamivir in patients with organ replacement therapy.

Another improvement would be to have more treatment options available in critically ill children with influenza infection. The European Centres for Disease Control and Prevention recommends NAI for the treatment of documented or suspected acute influenza A and B virus infection in persons who are hospitalized with influenza, outpatients with severe illness, outpatients who are at high risk of complications from influenza, children younger than 2 years and adults ≥ 65 years and pregnant women and those within 2 weeks postpartum [9,32,33,34,35]. The NAI’s mechanism of action is to prevent the release of replicated virus from infected cells [36,37]. There are four NAIs in use: zanamivir, oseltamivir, peramivir, and laninamivir. Zanamivir (4-guanidino-Neu5Ac2en), oseltamivir, and peramivir (DB06614) are derivatives of the earliest transition state analog of the NA substrate N-acetylneuraminic acid (2,3-dehydro- 2-deoxy-N-acetylneuraminic acid, DANA), whilst laninamivir (R125489) represents a 7-OCH3 substitution of zanamivir [37]. Two NAIs are licensed worldwide: zanamivir and oseltamivir phosphate (GS-4104) [38]. Peramivir is licensed in Japan, South Korea, and the USA, while laninamivir is only licensed in Japan but is currently in Phase III clinical trials in the USA [31,39]. Peramivir is a neuraminidase inhibitor that has a higher affinity for the influenza virus than oseltamivir [31]. In December 2014, peramivir was approved by the FDA for the treatment of acute, uncomplicated influenza in adults who have been symptomatic for ≤2 days [34]. In 2021, the FDA has approved the expanding of the patient population to include patients 6 months and older [35]. As of 2019 in Europe, intravenous zanamivir is approved for adults and children from 6 month of age with complicated influenza, that requires hospitalization of the patient, and when the virus is resistant to other influenza treatments or when other antiviral treatments, including inhaled zanamivir, are not suitable for the patient [9]. Before 2019, at the time of the patient’s illness, the compassionate use program had been discontinued and IV zanamivir compassionate use was the only IV treatment available in Europe.

Additional considerations include the combination of NAI with a second antiviral drug that exhibits a different mechanism of action, such as baloxavir, which has been licensed for influenza A and B viruses infections in patients > 12 years in Japan and United states since 2018 [40]. Baloxavir marboxil (S-033188) is the small-molecule prodrug of the selective polymerase acidic protein inhibitor S-033447 and inhibits the cap-dependent endonuclease and prevents viral transcription [41]. A study with 366 patients, 87% of whom had influenza A virus infection, combining baloxavir with NAIs (oseltamivir, zanamivir, and peramivir) did not result in superior clinical outcomes compared with NAIs alone [42]. In up to 9.7% of baloxavir recipients, PA I38T/M amino acid substitutions were detected [43].

Data on effective antiviral therapy for children with severe influenza virus infection are scarce. A retrospective Dutch study with two children among a total of thirteen patients showed limited effect of late add-on IV zanamivir therapy [44]. Fry et al. reported a US surveillance program in 2009/2010, where two children and six adults received IV zanamivir therapy, but the surveillance does not collect data to assess adverse events or the reasons for treatment and cannot evaluate the effectiveness of IV NAIs. The study indicates that most hospitalized patients with A(H1N1)pdm09 infection received treatment with an NAI; however, receipt of an experimental IV NAI was rare. Limited data on effectiveness and safety in severely ill patients and children and on dosing for renal failure were available for the IV NAIs. Patients who received experimental IV NAIs tended to be more severely ill and obese; thus, an adequate comparison group was not available [45]. Karsch et al. previously reported two patients (one pediatric, one adult) with severe influenza B virus infection requiring ECMO and organ replacement therapy. The study evaluated pharmacokinetics of oral and intravenous oseltamivir in different body compartments [23]. This study indicated that oral oseltamivir may be insufficiently absorbed in ICU patients, whereas IV oseltamivir showed pronounced effectiveness in depressing virus load. Capacity building for rapid-turnaround pharmacokinetics and pharmacodynamics should be considered to aid in the management of life-threatening influenza infection.

The virology analysis in the reported case is limited since the low viral load of the sample did not allow further NAI resistance analysis. Thus, it is unclear whether failure of oseltamivir therapy was due to resistance and/or malabsorption. In vitro data indicate that zanamivir treatment may still show efficacy in patients with the influenza resistance mutation [46,47]. Hussain et al. reported that a majority of oseltamivir- and peramivir-resistant influenza A viruses may still be largely sensitive to zanamivir and laninamivir. Of 21 tested mutations (V116A, E119A, E119V, Q136K, Q136K/R, N142S, D198E, I222K, I222R, I222T, S246N, H274Y, R292K, N294S, G320E, T156I + D213G, D198N + H274Y, I222R + H274Y, I222V + H274Y, I222T + S331R, S246N + H274Y) with 24 subtypes, 22 subtypes were resistant to oseltamivir or peramivir. Of these, 17 were sensitive to zanamivir or laninamivir. It was measured with the inhibition level (IC50 fold-change) [48].

We observed a drastic decline in myoglobin and creatinine kinase levels under IV zanamivir therapy—we assume that rhabdomyolysis was due to active influenza virus infection, and clinical and laboratory improvement were owed to the direct effect of antiviral therapy with IV zanamivir on virus replication.

Rhabdomyolysis in patients with influenza virus infection has been reported in the literature [10,11,12,13,14,15,16,17]. Patient management included aggressive fluid resuscitation, especially in one case of severe kidney failure requiring hemodialysis. This was a 28 year old woman, she received 22 l hydration over 96 h and 5 days of oseltamivir, 75 mg twice daily [10]. In our case, volume therapy, hemodialysis and IV zanamivir solved the rhabdomyolysis.

Creatinine kinase levels might be an important biomarker for rhabdomyolysis in patients with influenza virus infection. It could be an important indicator to determine whether the virus is active in muscle tissue, and whether antiviral therapy is effective [49]. Myoglobin levels may reflect the severity of disease in patients who develop rhabdomyolysis, as it has been found for COVID-19 patients [50].

The rapid improvement of rhabdomyolysis and the patient’s overall condition under IV zanamivir therapy underlines the need for licensed options for parenteral treatment in children with potentially life-threatening influenza virus infections.

## 4. Materials and Methods

### 4.1. Virology

The virology work and standardized clinical assessments were performed in the context of a Quality Management (QM) program at the Charité Department of Pediatrics (IRB approval number EA 24/008/10) in collaboration with the National Reference Centre for Influenza at the Robert Koch-Institute. Individual informed consent was obtained from this patient’s family permitting publication of a case report.

QM staff obtained nasopharyngeal specimens to be delivered to the Robert Koch-Institute in universal transport medium (CopanTM; Copan Diagnostics, Murrieta, CA, USA) for RT-qPCR analysis for influenza A(H3N2) using the same RT-qPCRs as in the national surveillance system [51]. BAL specimens were also delivered to the Robert Koch-Institute for the RT-qPCR analysis as mentioned above.

### 4.2. Routine Parameter Analysis

Myoglobin was analyzed in Labor Berlin by Immunoturbidimetry. The reference range was 19 to 51 µg/L for females and 23 to 72 µg/L for males [52].

Creatinine kinase was analyzed in Labor Berlin by a UV-Test. The reference range was up to 167 U/L for female and up to 190 U/L for male [53].

### 4.3. Clinical Severity Assessments

The ViVI Disease Severity Score is a 22-item weighed clinical composite score, according to WHO-criteria of uncomplicated and complicated disease. The ViVI Disease Severity Score is comprises nine items describing signs and symptoms of uncomplicated disease (disease severity, uncomplicated: DSU, weighed single-fold) reflecting ‘regular’ ILI activity, whereas the thirteen items describing parameters are consistent with complicated disease (disease severity, complicated: DSC, weighed threefold), indicating high-impact clinical presentations in the target population [20]. It was used as assessment tool constantly beside the laboratory and clinical findings and therapeutic procedures of ICU.

## 5. Conclusions

IV zanamivir is effective in severe influenza virus infection with rhabdomyolysis even when started more than 48 h after symptoms begin.

An adult dose of IV zanamivir could be considered in children with organ replacement therapy.

More studies of use in children over 6 months, especially in very severe cases, would be needed to better assess the efficacy and dosing of IV zanamivir.

## Figures and Tables

**Figure 1 pharmaceuticals-16-00085-f001:**
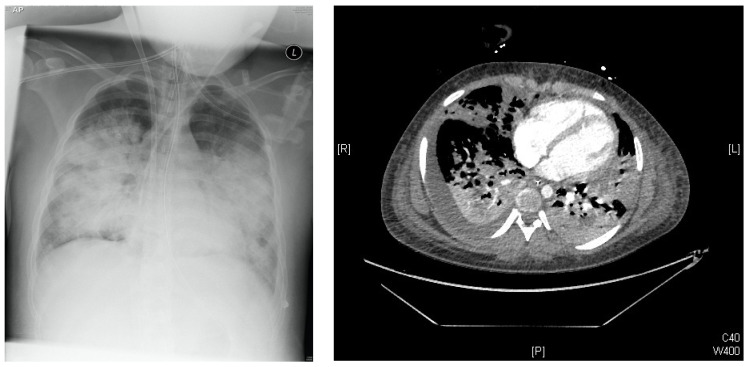
**Left**: Chest-X-Ray Day of illness #6 (DOI #6): Multiple bilateral pulmonary opacifications and lying ECMO cannula over the right internal jugular vein. **Right**: Chest-CT DOI#10: infiltrates on both sides with multiple pulmonary meltdowns and pleural effusions.

**Figure 2 pharmaceuticals-16-00085-f002:**
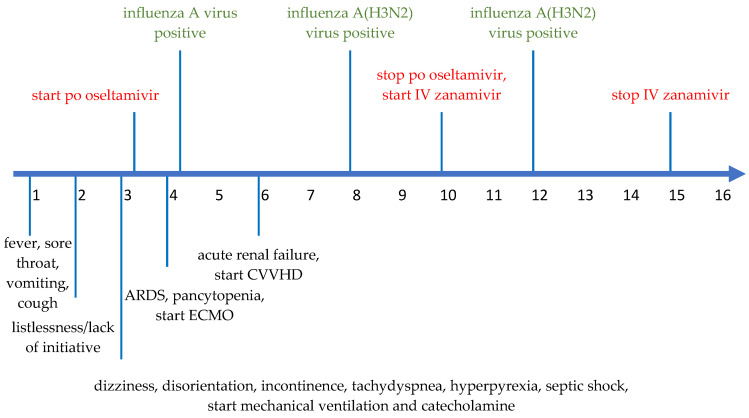
Timeline of disease progression from DOI#1 to DOI#16. The green items mean the virology data, the red items the application of po oseltamivir and IV zanamivir. The black items represent symptoms and treatment.

**Figure 3 pharmaceuticals-16-00085-f003:**
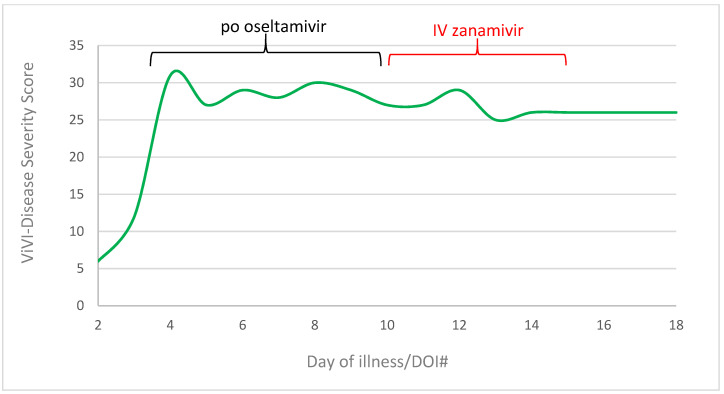
ViVI-Disease Severity Score in progress from day of Illness 2 to 18. Green line represents the disease severity score, the black line means the application of po oseltamivir (day 3 to 10), the red line the application of IV zanamivir (day 10 to 15).

**Figure 4 pharmaceuticals-16-00085-f004:**
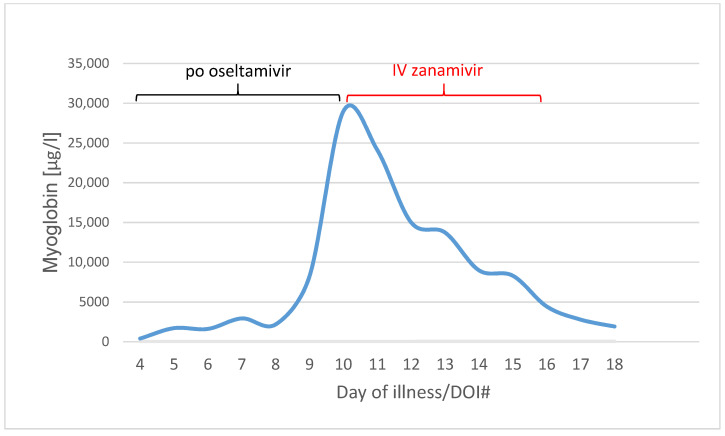
Myoglobin values in µg/L from Day of illness 4 to 18 (blue line). Black line means the po oseltamivir therapy (day 3 to 10), red line the IV zanamivir therapy (day 10 to 15).

**Figure 5 pharmaceuticals-16-00085-f005:**
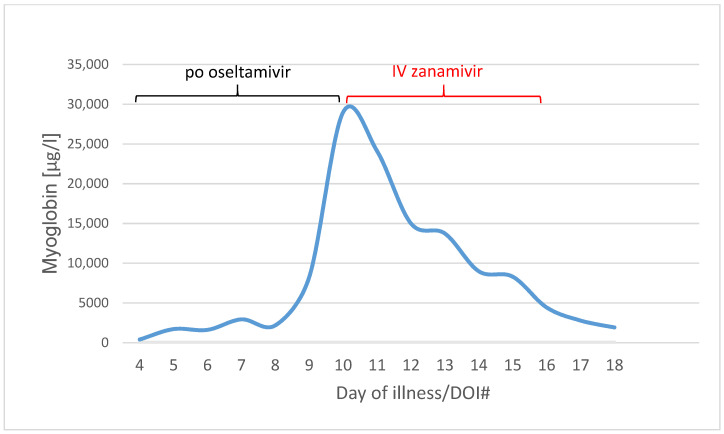
Creatine kinase values in U/L (violet line) from day of illness 4 to 18. Black line means the po oseltamivir therapy (day 3 to 10), red line the IV zanamivir therapy (day 10 to 15).

## Data Availability

Data is contained within the article.

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
