# Peer review of "Successful Treatment of Complicated Influenza A(H3N2) Virus Infection and Rhabdomyolysis with Compassionate Use of IV Zanamivir"

_pharmaceuticals, 2023, doi:10.3390/ph16010085_

Round 1
Reviewer 1 Report
The authors presented the successful use of IV zanamivir compassionate in a 14-year-old female with severe influenza A (H3N2) and multiple organ failure who had failed oral oseltamivir. However, no virus was detected by real-time RT-PCR on day 9 of illness (before zanamivir therapy has been stared) (p. 4, lane 129 – p. 5, lane 130). Therefore, it is considered that the influenza virus was cleared from the patient. Why did the authors add a different neuraminidase inhibitor in this situation? (Did the authors need additional medication (Neuraminidase inhibitor) ?)
Also, please explain what condition “failed a previous 7 day extended course of oral oseltamivir. (p. 2, lane 54 to 55)” indicates. As mentioned earlier, oseltamivir was effective against the virus, as virus was undetectable on day 9 of illness.
Author Response
Thank you for you comment. We are sorry if there was a misunderstanding.
The Zanamivir treatment started after Oseltamivir had failed.
The patient's clinical condition did not improve during oseltamivir therapy.
The fact that the virus was no longer detectable was not known at the time of the decision to start therapy with zanamivir (day 10).
We have now explained this circumstance in more detail in the paper (p. 2, lane 56).
Reviewer 2 Report
Alchikh and authors demonstrated the use of Zanamivir as a treatment option for Influenza A virus (H3N2). In this case study, authors have shown successful treatment in infected 14- year old patient with severe disease. The outcome of this treatment opens a possibility for patients with critical condition. I recommend acceptance of the article for publication.
Author Response
Thank you.Reviewer 3 Report
In this case report, Alchikh et al. reported a successful compassionate use of IV zanamivir in a 14-year-old female with severe influenza A(H3N2) and multi-organ failure. This clinical case report will guide the treatment of influenza viruses infection. The manuscript is well written and the data is well presented. A few comments below should be addressed.
1. I suggested the authors listed the disease progression by the graphic illustration.
2. figure 1 and figure 2 should be combined to one figure by different panels.
3. What is the meaning of “DOI#”? It appears sever times in the manuscript.
Author Response
Thank you for your valuable feedback
1. We have added a graphic with the disease progression.
2. We have combined it now to one figure.
3. Thank you. DOI# means Day of Illness. We have explained it now in line 70.
Round 2
Reviewer 1 Report
Regarding that point, I understand that there was a description in the content of the paper. However, the discussion (p.8, lane 258 – 261) gives the impression that zanamivir administration was effective after the virus had been cleared from the patient.
" We observed a drastic decline in myoglobin and creatinine kinase levels under IV zanamivir therapy - we assume that rhabdomyolysis was due to active influenza virus infection, and that clinical and laboratory improvement were owed to the direct effect of antiviral therapy with IV zanamivir on virus replication. "
What I would like to ask is how does author think about the mechanism of action of zanamivir? Does author think that zanamivir has effects other than antiviral effects? Is it possible that elimination of the virus from the patient by oseltamivir could improve other symptoms?
Author Response
Thank you for your comment. We apologize for this oversight.
After revisiting all virology results in our data base as well as the Robert Koch-Institute we can confirm that the latest Influenza A(H3N2) positive PCR was derived from a BAL sample from Day 12 of illness, with a ct value of 33. Additional ct values could not be determined due to negative results in all remaining samples.
The paragraph (page 4, line 78) was adjusted as follows:
“On day 8 of illness, the initial bronchoalveolar lavage (BAL) specimen was confirmed positive for influenza A(H3N2) by PCR at the National Reference for Influenza at the Robert Koch-Institute. Follow-up testing on day 12 was once again positive for Influenza A(H3N2) with a ct value of 33, follow-ups after this date remained negative. No other viruses were detected (tested for human rhinovirus, human respiratory syncytial virus, human adenovirus, human metapneumovirus).”
On page 5, line 152 we also adjusted the paragraph:
“Influenza A(H3N2) virus was detectable in BAL specimen on day 8 of illness (DOI#8) and on day 12 of illness (DOI#12, ct 33), after zanamivir therapy had been started. Virus load had become undetectable by RT-PCR on day 13 and remained undetectable thereafter.”
PO oseltamivir failed to eliminate the virus. After starting zanamivir the patient's condition stabilized and rapidly rising creatine kinase and myoglobin levels dropped immediately after initiation of IV zanamivir. Virus load became indetectable after day 2 of IV zanamivir. If the effect had been due to an extended 7-day course of oseltamivir, it should have occurred earlier.
Rhabdomyolysis is a known sign of progressive/severe and disseminated influenza virus infection, hence the most plausible explanation for the drastic clinical course reversal and improvement seen under IV zanamivir therapy is due to its improved bioavailability and its enhanced effect on virus load. As is well known, respiratory specimens only indicate a small part of the picture in other body compartments.
Round 3
Reviewer 1 Report
The authors have responded appropriately to my concerns, modified the manuscript.